# Does SARS-CoV-2 Trigger Stress-Induced Autoimmunity by Molecular Mimicry? A Hypothesis

**DOI:** 10.3390/jcm9072038

**Published:** 2020-06-29

**Authors:** Francesco Cappello, Antonella Marino Gammazza, Francesco Dieli, Everly Conway de Macario, Alberto JL Macario

**Affiliations:** 1Department of Biomedicine, Neuroscience and Advanced Diagnostics (BIND), University of Palermo, 90127 Palermo, Italy; antonella.marinogammazza@unipa.it (A.M.G.); francesco.dieli@unipa.it (F.D.); 2Euro-Mediterranean Institute of Science and Technology (IEMEST), 90141 Palermo, Italy; econwaydemacario@som.umaryland.edu; 3Department of Microbiology and Immunology, School of Medicine, University of Maryland at Baltimore-Institute of Marine and Environmental Technology (IMET), Baltimore, MD 21202, USA

**Keywords:** SARS-CoV-2, COVID-19, cell stress, antistress proteins, molecular chaperones, molecular mimicry, crossreactive antibodies

## Abstract

Viruses can generate molecular mimicry phenomena within their hosts. Why should severe acute respiratory syndrome coronavirus 2 (SARS-CoV-2) not be considered one of these? Information in this short review suggests that it might be so and, thus, encourages research aiming at testing this possibility. We propose, as a working hypothesis, that the virus induces antibodies and that some of them crossreact with host’s antigens, thus eliciting autoimmune phenomena with devasting consequences in various tissues and organs. If confirmed, by in vitro and in vivo tests, this could drive researchers to find effective treatments against the virus.

## 1. COVID-19 Disease: A Brief Overview

Severe acute respiratory syndrome coronavirus 2 (SARS-CoV-2) induced disease (COVID-19) is a planetary emergency that is urging many research groups to redirect their efforts and to channel their experience towards understanding its pathogenesis. Despite many clinical reports and papers on viral genetics, detailed information on pathogenic mechanisms pertaining to COVID-19 is still lacking. This type of information will no doubt help physicians in patient management and in providing treatment. The paucity of data on pathogenesis is due to a considerable extent to the very low number of autopsies that have been performed on COVID-19 victims [1]. While histopathological and other data from laboratory tests and autopsies will accumulate as the pandemic persists in the next few months or so, some progress can be achieved applying bioinformatics and scientific reasoning.

In this brief hypothesis paper, we have organized pertinent information available not only from the growing scientific literature but also from the chats of doctors and researchers on the web that cannot be ignored at this time, although they are not official instruments for dissemination of scientific data. These are temporarily useful channels for disclosing information as it is being generated at the “war front” (i.e., the doctors’ offices and clinical departments) that under normal circumstances would be available in the form of scientific publications only many months after the fact.

Among the numerous articles consulted, some have caught our attention [2,3,4,5,6,7,8,9,10,11]. By reading these and other publications, we arrived at the initial conclusion that COVID-19 develops in three steps (Figure 1 and Figure 2). In the following considerations, we will focus on the disease caused when the virus invades the body via the upper respiratory tract disregarding the other ways of viral entry, which are considerably less frequent as per current data—nevertheless, it is very likely that the conclusions would have also applied to the latter.

The first step consists of upper airway infection: the virus colonizes and multiplies in the ciliated columnar epithelial cells of the respiratory mucosa. This phase can be asymptomatic, paucisymptomatic, or symptomatic; in any case, an innate immune response against the virus is triggered. The disease can be resolved at this level (fortunately in most cases) or it can progress to the second step.

The second step is characterized by lung infection (bilateral interstitial pneumonia), which can be of varying severities. In the more fortunate cases, clinicians manage to contain the infection with antiviral and/or anti-inflammatory therapies (or the infection is self-limited, a possibility that cannot be excluded at this time). In more severe cases, for unknown reasons but which are probably related to a “hyperreactivity” of both innate and acquired immunities, the disease progresses towards the third step.

All the pieces of information available on the Internet agree in indicating that in the third step the disease is systemic (representative examples of clinical and laboratory studies are presented in Table 1). The emerging picture is that of widespread microvascular damage, diffuse thrombosis, disseminated intravascular coagulation (DIC) and, lastly, a multiorgan failure (MOF), often leading to death (Figure 3).

It is noteworthy that these patients do not show the typical features of disseminated intravascular coagulopathy (DIC). Typically, patients with DIC present with a considerably prolonged prothrombin time and a major reduction in platelet counts. By contrast, COVID-19 patients have a moderately prolonged prothrombin time and platelet counts are often in the lower range of the normal. This strongly indicates that blood-clotting activation in COVID-19 is different from the standard DIC clotting activation. Furthermore, the moderately reduced platelet count clearly resembles an immune complex-mediated prothrombotic disorder, e.g., heparin-induced thrombocytopenia [19].

Some reports of damage of extrapulmonary organs are listed in Table 2. In addition, at this writing there is growing evidence of autoimmune dermatitis, Guillain Barre syndrome, and Kawasaki disease in some COVID-19 patients, particularly the younger ones [20,21,22].

Mortality rate is very high if the patient is male, elderly, with other concomitant pathologies, especially those related to hypertension and/or diabetes [3,5,9,18].

Despite all the information summarized above, it is still a mystery what triggers the hyperactivation of the immune system, which is virtually always present. We have elaborated a working hypothesis [27] that we now would like to propose to the scientific community and, thus, provide food for thought and a basis to plan clinical and laboratory research.

## 2. Stress and Antistress Proteins

For many years our group has been studying a class of proteins highly conserved during evolution and organogenesis, the heat shock proteins (Hsp), many of which are molecular chaperones. These are typically antistress proteins (ASP) that have helped cells, since their origins at the beginning of life on earth, to survive environmental stresses of chemical and physical nature and have, therefore, played an important role in evolution [28]. Typically, ASPs are overexpressed in cells exposed to various kinds of stressors including bacterial and viral infections. Hsp/chaperones are essential for survival and maintenance of protein homeostasis in all organisms but, if abnormal can cause disease, the chaperonopathies [28]. Understanding the role of these proteins can provide novel elements for researchers and clinicians useful in diagnosis and treatment [29].

In the course of our studies, we came to the conclusion that Hsp/chaperones can be involved in molecular mimicry phenomena, most likely because of their long evolutionary history and high degree of structural conservation, which has produced a widespread sharing of various antigen within and across species. Hsp/chaperones are very similar in all organisms, from the most primitive unicellular to the most complex multicellular, typically sharing many highly immunogenic epitopes. This situation sets the stage for immunological crossreactivity between Hsp/chaperones from all over the spectrum of living organisms. For instance, Hsp/chaperones from any organism (bacterium, virus, or protozoan) in the human skin, or gastrointestinal, respiratory, and genitourinary tracts can invade the blood and thus come in contact with the host’s immune system. Antibodies are formed against the foreign Hsp/chaperone that will most likely crossreact with the equivalent molecule of the human host, and this would be a typical example of molecular mimicry [30]. The same can happen with other microbial and human molecules because there are epitopes shared not only between Hsp/chaperones but also between them and other molecules with different functions [31,32,33,34].

ASPs, including Hsp/chaperones, are typically intracellular molecules, but following stress events that augment their intracellular levels, they undergo post-translational modifications (PTM) and translocate to the plasma-cell membrane with their antigenic epitopes exteriorized on the cell’s surface [35,36,37,38]. These human epitopes, in turn, can be recognized by circulating antibodies made against crossreactive microbial antigens; these antibodies behave like autoantibodies, causing the destruction of the stressed cells, representing a typical example of pathology caused by molecular mimicry and manifested as autoimmunity [30].

Any ASP can be affected by PTM, which may change its properties and functions and make it pathogenic against its own host, generating a chaperonopathy by mistake [39,40].

We speculated that the progression of COVID-19 from step 2 to step 3 relies on molecular mimicry phenomena, as already shown for other viruses (Table 3).

## 3. Working Hypotheses

The severe bilateral pneumonia that develops in some individuals causes a drop in the partial pressure of oxygen in the blood. This undoubtedly represents a systemic stress. All cells suffer hypoxia, and this can lead to an overexpression of stress proteins and, in turn, to their change by PTM and translocation to the plasma cell-membrane. This would trigger molecular mimicry phenomena and a pathogenic cascade leading to MOF (Figure 3). It should be clear that this cascade can be triggered also by other crossreactive proteins distinct from ASP. Therefore, the search for the protein responsible for molecular mimicry cannot be limited to ASPs but must be extended to a wider range of cellular proteins. This search, now, is really a conundrum calling for concerted efforts of many research groups worldwide (Figure 4). It is important to bear in mind that, in addition to autoantibodies and their complexes with soluble or cell-surface antigens, other effectors of autoimmunity such as immunocompetent cells should also be sought for and characterized to obtain a comprehensive picture of the pathogenic mechanism underpinning tissue damage.

For example, one possible pathogenic mechanism of tissue damage is antibody dependent enhancement (ADE) of SARS-CoV-2 due to cross-reactivity. ADE has been recently claimed as a mechanism favoring Middle East Respiratory Syndrome Coronavirus (MERS-CoV) entry into host cells [51]. However, in a SARS-CoV macaque infection model, anti–spike IgG antibodies bind to the FcγR on alveolar macrophages and promote their activation with release of massive amounts of pro-inflammatory cytokines [52]. By analogy, anti-SARS-CoV-2/anti-ASP cross-reactive antibodies may similarly mediate ADE and contribute to tissue damage. This and other similar hypotheses need to be clarified.

To test the proposed working hypotheses, several steps are necessary, for example: (1) in silico comparison of epitopes of viral and human proteins, considering all these as putative autoantigens; (2) screening the results from the in silico studies, using the clues provided by epidemiological and clinical data being generated as the pandemic continues, to identify the protein(s) candidates; (3) immunohistochemical and other molecular analyses of tissues obtained from autopsies of COVID-19 fatalities to determine if, and where, these crossreactive molecules are expressed and are indeed reactive with pertinent antibodies.

What are the main clues that epidemiology and the clinics provide to date? They are many and disparate. We have listed some of them in Figure 4 and these can be classified into negative and positive prognostic factors.

In brief, from an epidemiological point of view [6], the main negative prognostic factors are the subject’s advanced age, the presence of comorbidities (hypertension and dysmetabolism), and the male sex. Conversely, main positive prognostic factors are young age (very few children are affected by the severe form of the disease), being female, and living in certain geographical areas. The latter might depend not only on the degree of environmental pollution or type of climate but also on genetic-driven protection that individuals might have developed by being exposed to other disease-causing agents.

Alongside these epidemiological indications, there are others that come from the clinic [3,4,5,8,11]. The disease, in the third step, involves endothelial cells and/or platelets and/or erythrocytes, more than other cells in the human body: this is suggested by signs of DIC and anemia often found by clinicians in SARS-CoV-2 infected patients. Furthermore, at this writing, it cannot be excluded that the renal failure that develops in many patients is not due either to the deposition of preformed circulating immune complexes or to the formation of immune complexes made by circulating antibodies bound to kidney antigens in situ. Last but not least, a critical examination of the molecular mechanisms underlying the efficacy of drugs that are currently being tested with some success as ex juvantibus therapy should not be overlooked, since it may offer further cues to unveil unknown pathogenic mechanisms.

All these clues, and others that may be revealed in the coming weeks from clinical and histological investigations, should guide researchers towards confirmation or exclusion of molecular mimicry as a determinant pathogenic factor.

## 4. Insights About SARS-CoV-2 Proteins

The understanding of SARS-CoV-2 phenotype using modern bioinformatics is critical to identify target proteins and shared epitopes between human and viral proteins. Here, we want to provide some preliminary insights.

Many research groups have described the virus by performing structural studies or extrapolating information available pertinent to other coronaviruses. By resorting to previously known information on genome sequences and protein structures and functions as well, bioinformaticians have been successfully assisting virologists by structurally characterizing proteins of the novel virus, determining the evolutionary trajectories, identifying interactions with host proteins, and providing other important biological insights.

The whole genome of SARS-CoV-2 was sequenced, and the sequence is available in GenBank (Accession number MN908947.3). Structural and nonstructural proteins were identified (12 reported in GenBank) and they are available in Protein Data Bank (PDB) database and The Universal Protein Resource (UniPROT). Moreover, many bioinformatics tools are currently used in the scientific literature to understand SARS-CoV-2 properties and in many cases are easily available online like Clustal Omega (EMBL-EBI, Cambridge, UK), BLAST, MODELLER, Mega-X, Swiss-Model (ExPASy, SIB Bioinformatics Resource Portal, Lausanne, Switzerland), just to mention a few examples. SARS-CoV-2 is a spherical or pleomorphic enveloped particle containing single-stranded RNA associated with a nucleoprotein within a capsid comprised of matrix protein. The envelope bears club-shaped glycoprotein projections [53]. SARS-CoV-2, as other coronaviruses, has four structural proteins, known as the S (spike), E (envelope), M (membrane), and N (nucleocapsid) proteins. Each of these proteins have a function since the N protein holds the RNA genome while the S, E, and M proteins together make the viral envelope [54]. The spike protein, which has been visualized at the atomic level using cryogenic electron microscopy [55,56], is the protein responsible for allowing the virus to attach to and fuse with the membrane of a host cell and for this reason it has captured major interest in the scientific community [54]. Experiments on the spike protein demonstrated that it has enough affinity to angiotensin converting enzyme 2 (ACE2) on human cells, supporting the idea that ACE2 is the cell entry receptor [10,55,56]. The S protein is composed of two functional subunits (S1 and S2) responsible for receptor binding and membrane fusion, respectively. The surface of the virally encoded envelope spike presents an array of host-derived glycans with each trimer displaying 66 N-linked glycosylation sites. This extensive glycosylation has important implications for vaccine design [57].

Recently, a comparative analysis of SARS-CoV-2 proteins with human proteins was performed in search of high local homologous matches [58]. Only one immunogenic epitope in SARS-CoV-2 had no homology to human proteins and it was concluded that, if all the parts of the epitopes that are homologous to human proteins are excluded from consideration due to risk of pathogenic priming, the remaining immunogenic parts of the epitopes may be still immunogenic and remain as potentially viable candidates for vaccine development. These results likely support our hypothesis and should prompt more investigations on this issue.

## 5. Conclusions

COVID-19 represents a global challenge for the medical community, researchers, and practitioners alike. We were not prepared from the health and social perspectives to face a pandemic, and states around the world are trying to adapt and find the best countermeasures and researchers are doing the same. At this moment, it is important not only to share the results, which may be few, but also the ideas, as these can serve as a stimulus to find solutions to the problems. With this short article, we wanted to offer our contribution, however small it might be, to face the challenge of the COVID-19 pandemic, stimulating the scientific community to investigate the involvement of molecular mimicry in the pathogenesis of COVID-19. This could be useful not only to reveal the pathogenetic mechanisms underpinning morbidity and mortality but also to direct the development of novel therapeutic strategies and a vaccine.

## Figures and Tables

**Figure 1 jcm-09-02038-f001:**
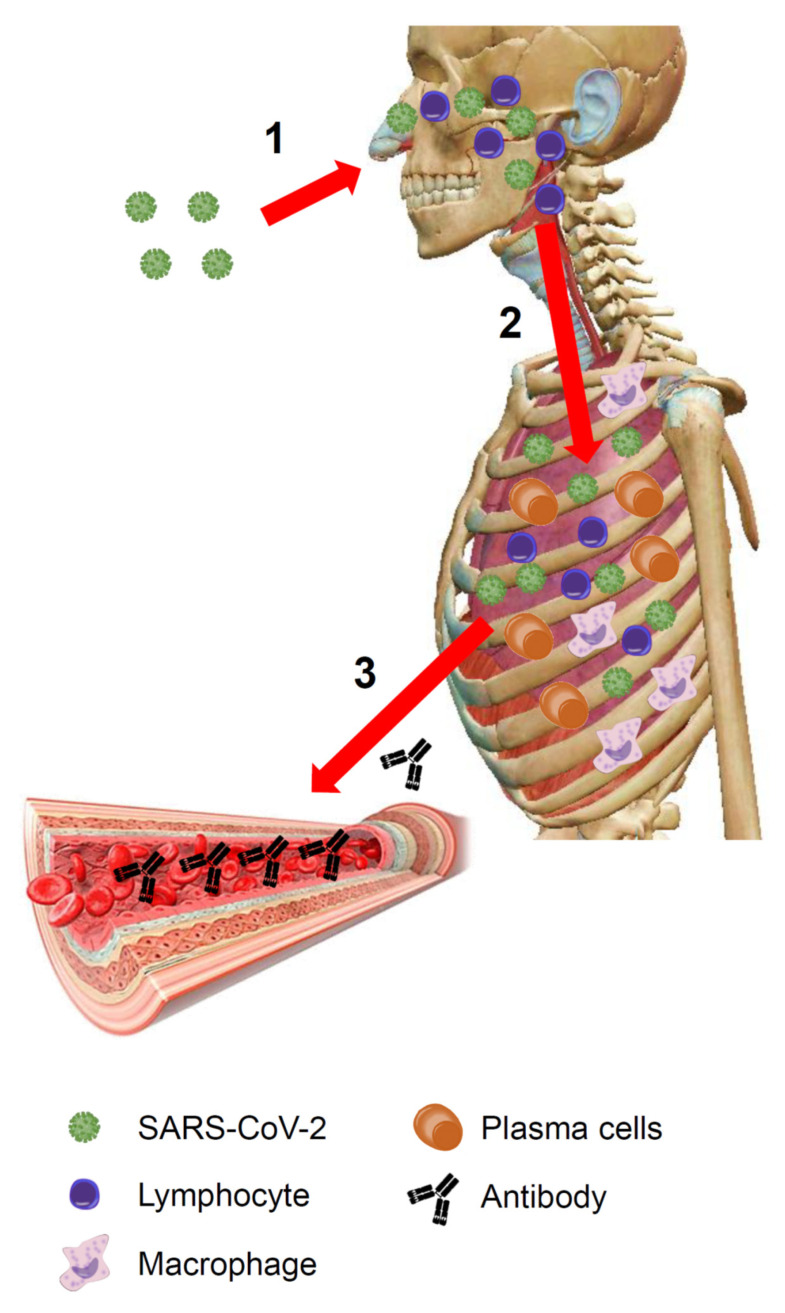
COVID-19: an overview. (**1**) The virus enters the body through the upper respiratory tract and invades the respiratory mucosa covering the nasal cavities, the paranasal sinuses, and the nasopharynx. Here it replicates and encounters immune cells. The immune system, via the Waldeyer’s ring, recognizes viral antigens activating innate immunity. (**2**) If the virus is not eradicated at this stage, it reaches the lower airways and enters the bloodstream through the respiratory barrier. The architecture of the primary pulmonary lobules is rapidly subverted by the violent inflammatory response, including both innate and adaptive immune-systems activation (lymphocytes, macrophages, plasma cells, etc.). (**3**) Plasma cells produce antibodies that by the bloodstream (the lung is a highly vascularized organ) can travel throughout the body. (The image of the human body is a courtesy of Visible Body Atlas.). SARS-CoV-2: severe acute respiratory syndrome coronavirus 2.

**Figure 2 jcm-09-02038-f002:**
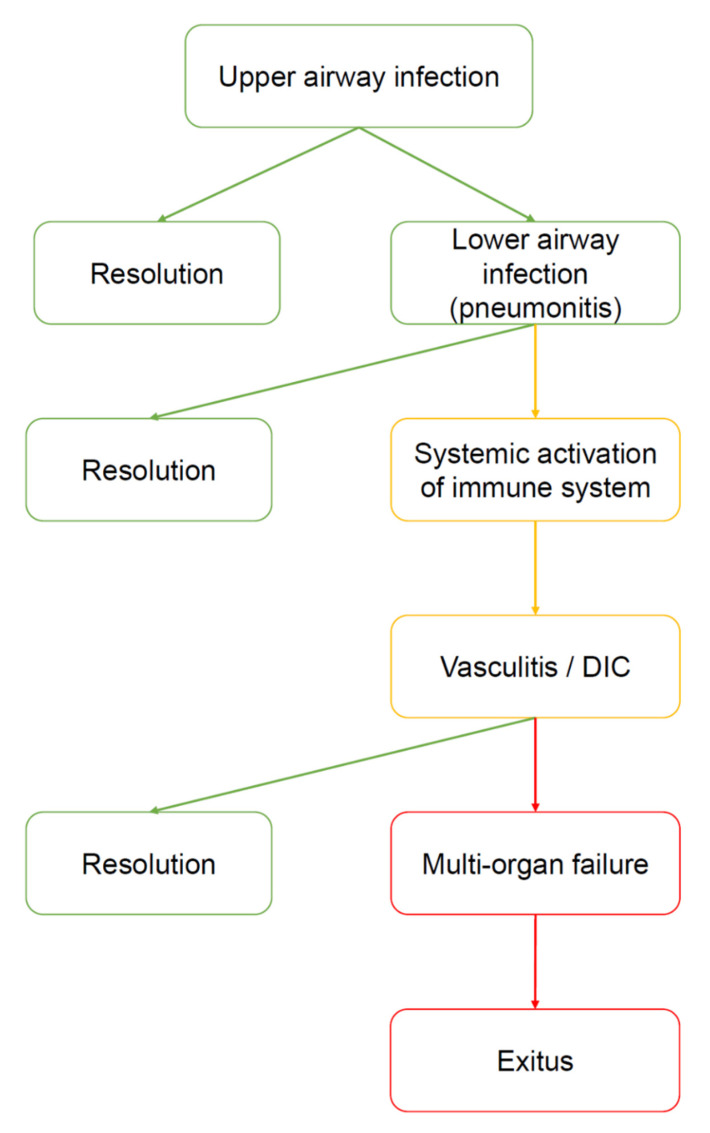
Natural history of COVID-19. The virus enters by the upper airways (nasal cavities). At this stage, the disease can be asymptomatic, paucisymptomatic or produce symptoms such as fever, cough, anosmia, ageusia, and shortness of breath. Many subjects heal spontaneously. However, in a limited number of subjects the virus moves down to the lower airways, causing severe pneumonia. It is not clear why some patients develop pneumonia and other do not. However, cold weather, high humidity, and severe pollution can be considered prodisease factors because they may favor virus vitality outside the body and inflammatory status inside the airways. Most of the patients with pneumonia manage to heal (for example, by ex juvantibus therapies, such as tocilizumab or hydroxychloroquine), however, some of them develop severe complications, i.e., a generalized activation of the immune system manifested as vasculitis, disseminated intravascular coagulation (DIC), and other signs and symptoms of autoimmunity. At this point, the risk of developing a multiorgan failure (MOF) is high, and the patient may die.

**Figure 3 jcm-09-02038-f003:**
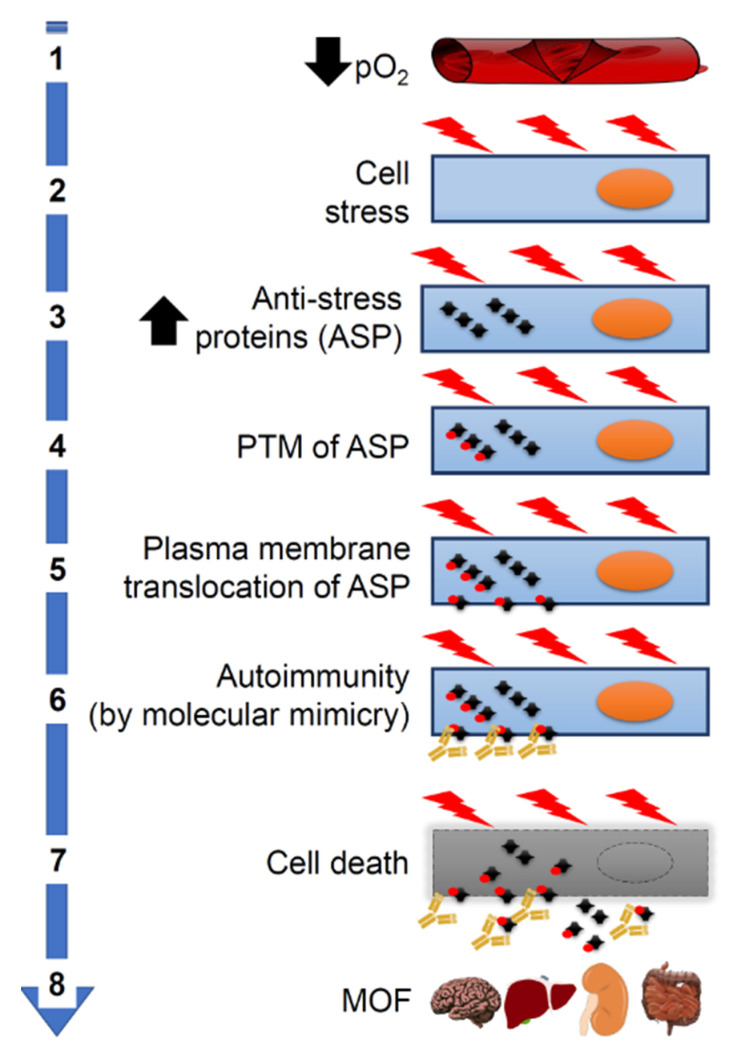
Pathogenesis of systemic complications of COVID-19. Molecular mimicry could be the cause of aggravation of COVID-19 patients through its participating in crucial steps of the pathogenetic cascade. (**1**) Severe pneumonia causes a decrease in partial oxygen pressure (pO₂). (**2**) This in turn causes cellular stress and increased synthesis of proteins (including antistress proteins, ASP). (**3**) ASP accumulate in the cytosol. (**4**) ASP undergo post-translational modifications (PTM). (**5**) Modified ASP migrate to the plasma cell membrane. (**6**) ASP antigenic epitopes shared with SARS-CoV-2 proteins (molecular mimicry) become accessible on the outer surface of the cells to crossreactive antiviral antibodies, which act as autoantibodies and cause autoimmunity. (**7**) Autoimmunity mechanisms damage and kill the host’s cells. (**8**) This kind of cell death occurs in many organs causing multiorgan failure (MOF).

**Figure 4 jcm-09-02038-f004:**
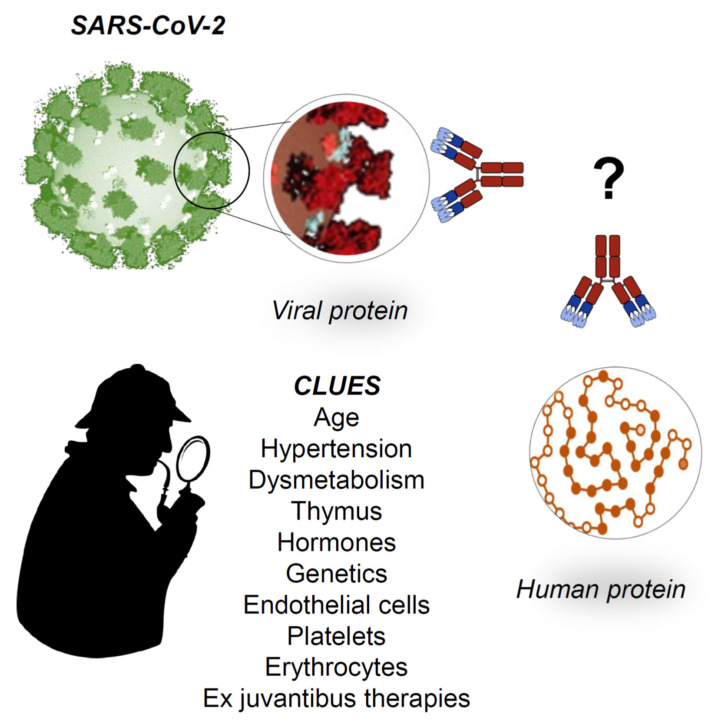
Working hypotheses. The establishment of generalized signs and symptoms of immune system activation indicates a serious aggravation of the COVID-19, which may be irreversible without proper medical intervention. Even with prompt medical intervention, the disease may follow its course and cause death. At the moment, there is no specific therapy for COVID-19, but clinicians use ex juvantibus therapy based on anti-inflammatory drugs such as tocilizumab (that inhibits IL6) and hydroxychloroquine (inhibits IL-1 and TNF-alfa); it is noteworthy that both drugs are used with success in autoimmune diseases. We hypothesize that, at the basis of the generalized activation of the immune system, there are molecular mimicry phenomena: the antibodies produced against the virus could turn into autoantibodies against crossreactive proteins expressed on human cells, causing autoimmunity with cell destruction. What proteins? Which cells? What are the predisposing factors? Furthermore, can there be protective factors? All of these are open questions now, although there are several clues that show directions for research in the immediate future.

**Table 1 jcm-09-02038-t001:** Examples of reports of generalized immune system activation in COVID-19.

Type of Study	Main Findings	References
Laboratory	Highlight the association between COVID-19 pathogenesis and excessive cytokine release from lungs, such as CCL2/MCP-1, CXCL10/IP-10, CCL3/MIP-1A, and CCL4/MIP1B.	[12]
Clinical and laboratory	Compared with nonintensive care unit (ICU) patients, ICU patients had higher plasma levels of interleukin (IL)2, IL7, IL10, GSCF, IP-10, MCP1, MIP1A, and TNFα.	[13]
Laboratory	SARS-CoV-2 infection significantly upregulated IL6, MCP-1, CXCL1, CXCL5, and CXLC10/IP10.	[14]
Clinical and laboratory	A retrospective multicenter study of 191 patients reported more elevated IL6 levels in nonsurvivors than in survivors; univariate analysis of the data revealed significant associations of elevated IL6 serum levels with mortality.	[2]
Clinical and laboratory	Compared to moderate cases, severe cases more frequently had dyspnea, lymphopenia, and hypoalbuminemia, with higher levels of alanine aminotransferase, lactate dehydrogenase, C-reactive protein, ferritin and D-dimer, as well as markedly higher levels of IL-2R, IL-6, IL-10, and TNF-α.	[15]
Clinical and laboratory	Elderly patients and with comorbidities showed higher plasma levels of IL6, IL10, lactate dehydrogenase, and c reactive protein.	[16]
Clinical and laboratory	Inflammatory cytokines were more elevated in severe cases than the nonsevere ones, including IL-2R, IL-6, IL-8, IL-10, and TNF-α. Immunoglobulins (Ig) A, IgG, and IgM and complement proteins (C3 and C4) in patients with COVID-19 were within normal range. There were no significant differences in the levels of IgA, IgG, and complement proteins C3 or C4 between the mild and severe groups, while IgM slightly decreased in severe ones.	[17]
Clinical and laboratory	Concentrations of alanine aminotransferase, aspartate aminotransferase, creatinine, creatine kinase, lactate dehydrogenase, cardiac troponin I, N-terminal probrain natriuretic peptide, and D-dimer were markedly higher in deceased patients than in recovered patients.	[18]

Notes. CCL2: chemokine (C-C motif) ligand 2; MCP-1: monocyte chemoattractant protein 1; CXCL10: C-X-C motif chemokine 10; IP-10: interferon gamma-induced protein 10; CCL3: chemokine (C-C motif) ligand 3; MIP-1A: macrophage inflammatory protein 1-alpha; GSCF: granulocyte colony-stimulating factor; TNFα: tumor necrosis factor alpha; CXCL1: C-X-C motif chemokine 1; CXCL5: C-X-C motif chemokine 5.

**Table 2 jcm-09-02038-t002:** Clinical and laboratory evidence of damage to extrapulmonary organs during SARS-CoV-2 infection.

Organ/System	Main Findings	References
Heart	Blood tests on admission showed most patients had higher levels of creatine kinase isoenzyme-myocardial band (CK-MB), myohemoglobin, cardiac troponin I, and N-terminal probrain natriuretic peptide.	[23]
Liver	COVID-19 patients had elevated levels of ALT, AST and bilirubin, respectively. ^1^	[24]
Kidney	On admission, 43.9% of patients had proteinuria and 26.7% had hematuria. The prevalence of elevated serum creatinine, elevated blood urea nitrogen, and estimated glomerular filtration under 60 mL/min/1.73 m^2^ were 14.4, 13.1, and 13.1%, respectively.	[25]
Nervous system	78/214 patients (36.4%) had neurologic manifestations including acute cerebrovascular diseases, impaired consciousness, and skeletal muscle injury.	[26]
Gastrointestinal tract	SARS-CoV-2 RNA was first detected in stool of the first reported COVID-19 case in the USA, who also presented with the digestive symptoms of nausea, vomiting, and diarrhea.	[4]

¹ Abbreviations: ALT: alanine aminotransferase; AST: aspartate aminotransferase.

**Table 3 jcm-09-02038-t003:** Examples of molecular mimicry involving viruses in disease.

Virus	Main Findings	References
Alphavirus	Sequence alignment of structural polyproteins belonging to arthritogenic alphaviruses revealed conserved regions which share homology with human proteins implicated in rheumatoid arthritis.	[41]
Cytomegalovirus	Human antibodies against UL44 (an obligate nuclear-resident, nonstructural viral protein vital for human cytomegalovirus (HCMV) DNA replication) immunoprecipitated nuclear-resident systemic lupus erythematosus autoantigens (namely, nucleolin, dsDNA, and ku70).	[42]
Coronaviruses	Several T-cell lines isolated from multiple sclerosis patients showed cross-reactivity between myelin and coronavirus antigens.	[43]
Enterovirus	Immunogenic epitopes in enterovirus capsid protein VP1 and procapsid protein VP0 have sequence similarities with diabetes-associated epitopes in tyrosine phosphatase IA-2/IAR and heat shock protein 60.	[44]
Epstein-Barr virus	Anti-C1q in systemic lupus erythematosus (SLE) patients can be induced by an EBV-derived epitope through molecular mimicry.	[45]
Papillomavirus	A potential antigenic mimicry between viral and human proteins may be causative of myalgic encephalomyelitis and chronic fatigue syndrome.	[46]
Rotavirus	In active celiac disease, a subset of antitransglutaminase IgA antibodies recognize the viral protein VP-7, suggesting a possible involvement of molecular mimicry in the pathogenesis of the disease.	[47]
Varicella-zoster virus	Autoantibodies to protein S can induce vasculitis and direct endothelial damage.	[48]
West Nile Virus	An in-silico analysis unveiled certain sequence similarities between viral antigens and receptor sequence fragments suggesting a molecular mimicry autoimmunization process.	[49]
Zika and dengue viruses	Anti-non-structural protein 1 antibodies can cross-react with host platelets and endothelial cells.	[50]

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
