# Peer review of "Does SARS-CoV-2 Trigger Stress-Induced Autoimmunity by Molecular Mimicry? A Hypothesis"

_jcm, 2020, doi:10.3390/jcm9072038_

Round 1

Reviewer 1 Report

In the revised version of the authors have well responded to my comments. Thank you for providing these interesting concept.

Author Response

We want to thank very much the reviewer for the positive answer!

Reviewer 2 Report

Well done.  Check for double spaces and typos.

Author Response

We thank the reviewer for the positive comment! We checked, found and corrected two typos and three double spaces.

Reviewer 3 Report

The paper “Does SARS-CoV-2 trigger stress-induced molecular mimicry pathogenic phenomena? A hypothesis” by Francesco Cappello et alpresents a hypothesis that infection by SARS-CoV-2 can induce autoimmune response and such response contributes to the pathogenesis of SARS-CoV-2 by causing devastating damages in various tissues and organs. The paper begins with a brief review of recently published literatures summarising the progression of the COVID-19 disease, by the end of the first section, the authors stated that it has been reported that many COVID-19 cases show evidences of immune system hyperactivation. Based on this observation, authors went on to propose immune hyperactivation is potentially a result of autoimmunity against cellular proteins that is released by cellular stress and damage during virus infection. The authors proposed that sequence similarity between SARS-CoV-2 proteins and host proteins led to generation of autoimmune antibodies that lead to further tissue and organ damage during virus infection.

This review/hypothesis paper is well written and well referenced overall. The proposed hypothesis is sound and I agree with the authors that pathogenesis by autoimmunity during coronavirus infection is an interesting area to investigate. There are only a few minor points that have brought to my attention.

1.

I found the title is rather confusing and not easy to understand without reading the text, there are a lot of focus in the paper on autoimmunity, I guess in the title “autoimmunity” is rephrased as “molecular mimicry pathogenic phenomena”, this is not straight forward to understand, and I would recommend to change the title to something like “Does SARS-CoV-2 trigger stress-induced autoimmunity by molecular mimicry”

2.

From line 71 to line 81, there are two paragraphs summarizing the first two stages of COVID-19 disease, it is not clear these are authors own clinical summary or based on findings published in literature, if it is the latter, events described in these two stages should be properly referenced.

3.

Line 242, “visualized at the atomic level using cryogenic electron microscopy [35].” Here, apart from ref 35, authors should also cite “Structure, Function, and Antigenicity of the SARS-CoV-2 Spike Glycoprotein, by Walls et al published in Cell

4.

Line 233-246 “Protein modeling experiments on the spike protein demonstrated that it has enough affinity to the receptor angiotensin converting enzyme 2 on human cells using them as a mechanism of cell entry [36].” This inappropriate citation, that besides modeling, a number of studies have presented direct experimental evidence that there is high affinity between Spike and ACE2, including reference 35 and the above mentioned reference which should be cited together with ref 35. In Addition, Ref 10 in the paper has shown the structure of S forming a complex with ACE2, supporting the idea that ACE2 is the receptor for SARS-CoV-2. Also consider rephrase the sentence to “Experiments on the spike protein demonstrated that it has enough affinity to angiotensin converting enzyme 2 on human cells supporting the idea that ACE2 is the cell entry receptor” Cell entry by coronavirus is a complicated process, involving receptor binding, protease cleavage and membrane fusion, binding receptor alone is not the mechanism of cell entry.

With these points appropriately addressed I would like to support the publication of the paper.

Author Response

We want to thank the reviewer for these positive comments and constructive remarks. We addressed all the points. Particularly:

  1. We modified the title as suggested by the Reviewer.
  2. We did not add any reference here because it is the authors own clinical summary, as guessed by the Reviewer.
  3. We added the reference suggested by the Reviewer.
  4. We rephrased the sentence as suggested by the Reviewer, adding the appropriate references and deleting the inappropriate one.

All the changes were made by using the "track change".

We hope the Reviewer will appreciate our changes and we want to thank again him/her for the help to improve our paper!